# A Comparative Analysis of Robot-Assisted Laparoscopic Pyeloplasty in Pediatric and Adult Patients: Does Age Matter?

**DOI:** 10.3390/jcm11195651

**Published:** 2022-09-25

**Authors:** Bosik Kang, Jungyo Suh, Bumjin Lim, Kun Suk Kim, Sang Hoon Song

**Affiliations:** Department of Urology, Asan Medical Center, University of Ulsan College of Medicine, Seoul 05505, Korea

**Keywords:** ureteropelvic junction obstruction, robotic pyeloplasty, urolithiasis

## Abstract

We investigated factors that affect the surgical outcomes of robotic pyeloplasty by comparing the surgical results of pediatric and adult patients with ureteropelvic junction stricture (UPJO). We retrospectively reviewed patients who underwent robotic pyeloplasty for UPJO between January 2013 and February 2022. The patients were categorized into two groups: the pediatric (≤18 years) and adult (>18 years) groups. The perioperative and postoperative outcomes and surgical complications were comparatively analyzed. Prognostic factors for predicting surgical failure were analyzed with multivariable logistic regression analysis. The pediatric group showed longer total operation and console times. The mean pain score was lower in the pediatric group than in the adult group on days 1 and 2 after surgery. The average amount of morphine used in the pediatric group was lower during postoperative days 0–2. No differences in the length of hospital stay, incidence of surgical failure, and incidence of urolithiasis requiring treatment after robotic pyeloplasty were observed between the groups. The only factor that predicted surgical failure was a history of urolithiasis before surgery. The results showed that age did not affect the surgical outcome.

## 1. Introduction

Ureteropelvic junction obstruction (UPJO) blocks urine flow from the renal pelvis to the ureter. Most cases of UPJO are congenital, and if inappropriately treated, it may result in hydronephrosis, chronic infection, and progressive loss of renal function [1,2]. Although the probability of spontaneous improvement of hydronephrosis due to UPJO without the need for surgery is nonnegligible, evaluating the necessity of surgical intervention for individuals who have a high risk of renal function deterioration is important [3,4].

Until recently, open pyeloplasty has been the main surgical option. However, in the last 20 years, less invasive surgical methods have been developed and popularized [5]. Not only laparoscopic pyeloplasty (LP) but also robot-assisted laparoscopic pyeloplasty (RALP) have been widely adopted as surgical options for treating UPJO [6]. Compared with LP, RALP has clear advantages, as follows: it enables 3D visualization and strengthens dexterity; it provides a more comfortable environment for surgeons; patients show shorter hospital stays with reduced narcotic requirements [7,8,9].

The introduction of robotic surgery in children has been slower, although it showed promising results [10,11]. RALP in pediatric patients has been reported to be associated with shortened operating hours, shortened postoperative hospital stays, low complication rates, and high surgical success rates [12,13,14]. Recent studies even suggest that age is not even considered a factor when deciding the need for robotic surgical intervention because significant benefits with RALP can be generally expected [15]. However, although such radical proposals are constantly being suggested, research comparing the outcomes of RALP between children and adults remains insufficient [12,13,14]. Therefore, in this study, we assessed the outcomes of robotic pyeloplasty performed in children and adults and examined whether age should be considered an influencing factor for surgical indication.

## 2. Materials and Methods

### 2.1. Patient Selection

We retrospectively reviewed the medical records of patients who underwent robotic pyeloplasty for UPJO at the Asan Medical Center between January 2013 and February 2022. The diagnosis of UPJO was based on clinical symptoms and imaging studies, such as renal ultrasonography (US) and Tc-99 m mercaptoacetyltriglycine (MAG3) renal scans. Surgical treatment was indicated when the patient had symptoms, such as abdominal or flank pain, progressive hydronephrosis, and renal functional deterioration. In this study, 117 patients with UPJO who underwent robotic pyeloplasty were enrolled and divided into two groups according to age: the pediatric (≤18 years) and adult (>18 years) groups. The study design and the use of patient data were approved by the Institutional Review Board of the Asan Medical Center (2021-1255).

### 2.2. Clinical Variables

The medical records of the 117 patients were reviewed for their hospital course and their perioperative data, as follows: patient demographic characteristics; perioperative data, such as total operation time (from the start of anesthesia to extubation), console time, anastomosis time, postoperative split renal function (SRF) results, postoperative serum creatinine levels, pyeloplasty methods, and surgical approaches; the length of hospital stay; postoperative daily pain scores; the total amount of analgesics used, which was converted into morphine equivalent doses.

### 2.3. Surgical Techniques and Postoperative Management

RALP was performed with the da Vinci S, Si, X or Xi Intuitive Surgical System robot according to previously reported techniques [14]. The patient was positioned in a modified lateral position. An 8 or 12 mm camera port was placed above umbilicus and positioned leaving a distance of at least 6 cm from each trocar or equidistant from the renal pelvis and the camera port. In most patients, an additional 5 or 12 mm assistant port was placed in patients at the suprapubic area for assistance with placement, suturing, suction or drainage. Robot arm port placement was different between the right and left side surgery (Figure 1 and Figure 2). An 8 mm port was most commonly used, while a 5 mm port was also used in few patients. Transmesenteric access to the retroperitoneum was used whenever possible. Renal pelvis and proximal ureter were dissected and mobilized. If necessary, a vascular hitch was performed. To expose the renal pelvis, a 2-0 nylon suture for extracorporeal knot tying was used if needed. Pyelotomy was performed with redundant pelvis tissue used to aid in anastomosis and suturing in order to protect the anastomosis site as much as possible. Continuous 5-0 polyglyconate and interrupted 5-0 or 6-0 polyglactin sutures were used for posterior layer of anastomosis. To prevent obstruction of anastomosis, double-pigtail stent was introduced via assistant port with a guidewire. When the assistant port was not placed, a 16-gauge angiocatheter was passed through the anterior abdominal wall near the hitch stitch. The stent with guidewire was passed through the angiocatheter and into the ureter. To confirm its position, the bladder was filled with saline mixed with indigo carmine. An anterior wall anastomosis was completed after stent positioning (Figure 3). After the operation, we prescribed morphine to patients with a numerical pain rating scale score of ≥4, following the perioperative pain management guidelines of our institution. The Face, Legs, Activity, Cry, Consolability behavioral pain scale was applied to pediatric patients to measure the degree of pain. The stent was removed 4–6 weeks postoperatively in an outpatient clinic using sedative medicine, such as ketamine and midazolam. Postoperative kidney US and a Tc-99 m MAG3 renal scan (MAG3) were performed 3, 6, and 12 months after surgery and annually thereafter. Surgical failure was defined as evidence of increased hydronephrosis on follow-up US or decreased differential renal function of >10% on follow-up MAG3 within 36 months or cases that underwent redo pyeloplasty after robotic pyeloplasty.

### 2.4. Statistical Analysis

Continuous variables were summarized as medians. Categorical variables were summarized as frequency counts and percentages. Differences between the groups were investigated using the t-test and chi-square test. Multivariable logistic regression analysis was performed to investigate the overall causes of surgical failure. *p*-values of less than 0.05 were used to denote statistical significance. All statistical analyses were performed using Statistical Package for the Social Sciences, version 26 (IBM Corp., Armonk, NY, USA).

## 3. Results

The characteristics of the 117 patients stratified by age are summarized in Table 1. In total, the pediatric group consisted of 48 patients, and the adult group comprised 69 patients. The mean age was 7.5 years in the pediatric group and 39.3 years in the adult group. The pediatric group comprised more male patients (70.8%) than the adult group (44.9%). In both groups, the left side was more commonly affected, and the most frequent initial presentation was flank/abdominal pain. The transmesenteric surgical approach was more commonly feasible in the pediatric group (45.8%) than in the adult group (23.2%).

Comparing perioperative and postoperative outcomes, the pediatric group showed a longer total operation and console times; however, no significant difference in anastomosis time was observed between the two groups (Table 2). However, when subdividing the pediatric group into infants (<1 year) and older pediatric patients, the total operation time was shorter in infant patients (mean, 134 min) than in older pediatric patients (mean, 176 min) (Appendix A). When the etiology was categorized into intrinsic (primary and polyps) and extrinsic (crossing vessel) causes, no significant difference in the distribution of etiologies was observed between the two groups. Moreover, postoperative SRF and serum creatinine did not show significant differences between the groups. The pediatric group showed lower pain scores on the day after surgery than the adult group. Additionally, the pediatric group showed a lower morphine equivalent dose of analgesics used from the day of surgery to the second day after surgery.

The results of the comparison of the surgical outcomes between the two groups are shown in Table 3. The surgical complication with the Clavien–Dindo classification grade of ≥3 was demonstrated. There were three postoperative complication cases in the pediatric group, two cases in the adult group and two surgical failure cases in both groups, respectively. Nevertheless, no significant difference in the surgical failure rate was observed between the two groups. In six infant patients, no surgical failure or postoperative urolithiasis was observed (Appendix A). Three patients in the pediatric group and two patients in the adult group had urolithiasis after the surgery. Two pediatric patients underwent extracorporeal shock wave lithotripsy (ESWL) and the other pediatric patient underwent RIRS. Meanwhile, one adult patient who developed urolithiasis after pyeloplasty underwent ESWL, and the other adult patient underwent RIRS. Still, no statistically significant difference in the occurrence of urolithiasis after surgery was observed between the two groups (*p* = 0.400).

There were three postoperative complication cases in the pediatric group and two cases in the adult group, respectively. The three complication cases in the pediatric group were as follows: (1) one underwent an antegrade pyelogram to remove percutaneous nephrostomy after surgery; however, the contrast medium failed to pass through the urinary tract. The patient was discharged after double-J (DJ) stent insertion. (2) One underwent ureteroscopy due to failure of DJ stent removal. (3) One underwent DJ stent reposition due to ureterovesical junction stricture-induced DJ stent malposition. The two complication cases in the adult group were as follows: (1) one was treated in the intensive care unit after the surgery because of the postoperative idiopathic hypoxic event. (2) One underwent RIRS due to the recurrence of a renal stone, which was initially removed during surgery.

There were two surgical failure cases in both groups, respectively. In the pediatric group, one patient was found to have aggravated hydronephrosis, and the other showed decreased SRF at 36 months. In the adult group, one patient was found to have aggravated hydronephrosis, and the other underwent reoperation because of massive burden of renal stone. To further investigate the overall causes of surgical failure, a logistic regression analysis was performed (Table 4). We performed univariate and multivariable logistic regression analyses to determine the potential variables that predict surgical failure after robotic pyeloplasty. Obesity, a history of urinary stone(s) before surgery, age, anteroposterior pelvic diameter, and SRF (<30%) were included in our model to predict surgical failure. Among these, obesity (odds ratio [OR], 10.6; *p* = 0.045) and the existence of stone(s) before surgery (OR, 14.8; *p* = 0.022) were significantly associated with surgical failure in the univariate analysis. In contrast, age, anteroposterior pelvic diameter, and SRF were not associated with surgical failure. When the multivariable analysis was performed, only a history of urinary stone(s) was found to be a significant factor for surgical failure (OR, 14.5; *p* = 0.022).

## 4. Discussion

This study compared the outcomes of robotic pyeloplasty according to age. To the best of our knowledge, this study has the largest number of patients among comparative studies on pediatric and adult pyeloplasty published until now. We demonstrated the efficacy and safety of RALP not only in adult patients but also in pediatric patients, even in infantile patients.

This study showed a high surgical success rate in both the pediatric and adult groups. The successful outcomes of RALP in pediatric and adult patients have been consistently reported in the literature worldwide. Minnillo et al. reported that the success rate of RALP in pediatric patients is 96% [16]. One recent meta-analysis indicated that robotic pyeloplasty is successful in 95.4% of pediatric patients [17]. Even when the scope of the subjects is narrowed down to ages lower than 1 year, it still shows a high success rate of 96% [18]. In adult patients, excellent outcomes have been reported, even when a larger number of cases are examined [19]. In a multicenter study, Mufarrij et al. reported that the rates of radiographic resolution of obstruction, major complications, and minor complications in patients with a mean age of 38.5 years were 95.7%, 7.1%, and 2.9%, respectively [20]. However, a direct comparison of the surgical outcomes between pediatric and adult patients with details on the analgesics used and complication profiles performed by a single surgeon is still lacking. This study confirmed the safety and efficacy of RALP implementing the same surgical techniques in any age group.

The time spent during a surgical procedure may reflect its complexity or level of difficulty. In this study, the total operative and console times were longer in the pediatric group. A smaller anteroposterior peritoneal diameter and a smaller caliber of ureteral diameter in children made the surgeon move slower and use the motion scaling option of the robotic system. However, after overcoming the learning curve, the surgeon could move efficiently even in infant patients, and this was demonstrated by a shorter operation time in infant patients than in adult and older pediatric patients (Appendix A).

In previous studies, RALP was shown to have an advantage over open pyeloplasty in terms of the use of pain medication [7,8]. Lee et al. reported that the total narcotic medication dosage was significantly less in 33 patients with a mean age of 7.8 years undergoing RALP than in 33 patients with a mean age of 7.6 years undergoing open pyeloplasty [7]. In a comparative study between pediatric and adult patients who underwent RALP, Mizuno et al. showed that the surgical outcomes were favorable and safe for both pediatric and adult patients; however, the analgesic dosage was not compared between both groups [21]. In contrast, we explored the amount of pain medication and pain scores in detail every day after surgery during the entire hospital stay. The pain scores on the day after surgery were lower in the pediatric group, and they showed lower morphine demand from the day of surgery to the second day after surgery. Although patients are known to have the most pain within 48 h after surgery [22], children who are undergoing RALP seem to suffer from pain less than adults, even in these earlier postoperative periods.

Three (2.5%) pediatric patients had urinary tract infections (UTIs) within 30 days after surgery. These patients kept their DJ stents for >4 weeks without taking prophylactic antibiotics. The incidence of postoperative UTI after pyeloplasty was reported to range from 2% to 15% [23]. Because urine is easy to reflux when a DJ stent exists, UTIs are thought to originate from prolonged DJ stent maintenance [24]. After removing the DJ stent within 2 weeks after surgery, no further UTIs have occurred. Therefore, we agree with Vidovic et al. who reported that a routine administration of prophylactic antibiotics after pyeloplasty does not appear to be beneficial [25].

We demonstrated that a history of urinary stones before surgery and obesity were significantly associated with the failure of RALP. Nayyar et al. speculated that stones in the collecting system proximal to the UPJO complicate the situation by inducing infection or inflammation, which makes the tissues edematous and friable [26]. In contrast, in a comparative study of LP with and without concomitant pyelolithotomy, Kadihasanoglu et al. reported similar success rates of 92.9% and 93.3%, respectively [27]. However, their study defined surgical success based on a negative renal scan only 3 months after surgery. In contrast, we defined surgical failure based on a longer follow-up duration up to 36 months. Our results support the statement of Chow et al. that the stone burden and the risk of postoperative stone formation should be considered possible postoperative issues [28].

Crossing vessels account for 20–40% of UPJO in all age groups [29,30]. In this study, it accounted for 27.1% of pediatric UPJO and 43.5% of adult UPJO. Although the difference was not statistically significant, a slightly higher prevalence of crossing vessels than in previous open series studies might be owing to the increased field of view around the UPJ area and increased chance of diagnosing one through the laparoscopic transperitoneal approach. Nevertheless, crossing vessels were not found to be a predictive factor for surgical failure in this study. We speculate that RALP enabled the surgeon to delicately dissect the renal pelvis without injury to major vessels and even crossing vessels and reposition the UPJ to the top of the crossing vessel to achieve a successful anastomosis in all patients.

The transmesenteric approach could be applied to fewer patients in the adult group (23.2%) than in the pediatric group (45.8%). Children have relatively less mesenteric fat, making it easier to access using the transmesenteric approach [19]. When the subjects in this study were divided into the transmesenteric approach group (*n* = 38) and nontransmesenteric approach group (*n* = 79), the operation time was shorter in the transmesenteric approach group than in the nontransmesenteric approach group (144 min vs. 154 min, *p* = 0.045). It was confirmed that the operation time could be reduced through the transmesenteric approach; however, it did not affect the surgical success rate.

Five patients could be operated on with RALP without a 5 mm assistant port. Not using an assistant port has the advantage of reducing one surgical scar; however, the operation time was prolonged, since the access port makes it possible to suction urine spillage and blood and to deliver surgical devices quickly. To achieve desirable cosmetic outcomes, when using an assistant port, we placed the port in the suprapubic area below the belt line, and a second robot arm port was placed in the lower abdomen below the anterior superior iliac spine to maximally hide the scars down the beltline. Although a visible scar is formed at the subxiphoid port site, it usually faints out to be rarely seen after a few years (Figure 4) [14].

This study has some limitations that should be acknowledged. First, this was a retrospective analysis with a limited number of patients. Further research is needed with sufficient cases of pediatric and adult patients. Moreover, this study cohort included the initial 20 patients who underwent RALP during the learning curve period of a surgeon. Studies suggested that 20–40 surgeries are needed to reach surgical proficiency for robot-assisted laparoscopic surgery [31,32]. This explains the little longer surgical time in older pediatric patients than in infants in our study cohort. Most infant cases in this study were operated on after overcoming the learning curve of robot surgery after >20 cases. Nevertheless, note that this study demonstrated that robotic pyeloplasty is highly successful, and the incidence of complications is low in both pediatric and adult patients.

## 5. Conclusions

When comparing the outcomes of robotic pyeloplasty between the pediatric and adult groups, the two groups showed no difference in anastomosis time. Children undergoing RALP suffer from pain less than adults, even in earlier postoperative periods. No significant difference in the postoperative complication rate or surgical failure rate was observed between the two groups. A history of urolithiasis before robotic pyeloplasty was the only predictor of surgical failure. We suggest that age is not a limiting factor for indicating robotic pyeloplasty in patients with UPJO.

## Figures and Tables

**Figure 1 jcm-11-05651-f001:**
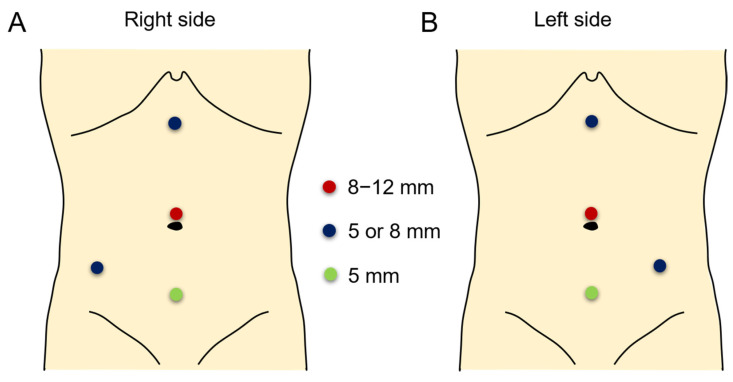
Port placement in right side (**A**) and left side (**B**) surgeries. The camera port was placed in the supraumbilical area with an 8, 8.5, or 12 mm port (red circle). The robotic arm ports were placed at the subxiphoid and lower abdominal areas (blue circles). An assistant port was placed at the suprapubic area (green circle).

**Figure 2 jcm-11-05651-f002:**
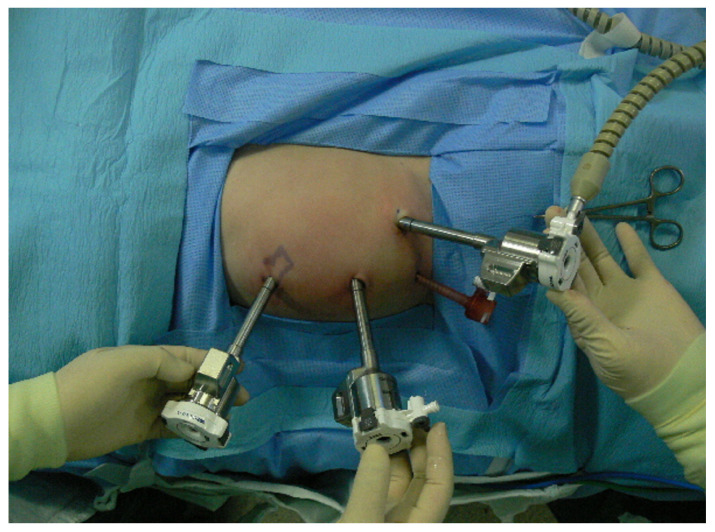
Port placement in 10-months-old infant with ureteropelvic junction obstruction on the left side. Three 8 mm robotic arm ports are placed in the abdomen. Subxiphoid and left lower abdominal areas are the robotic arm ports for left and right hands. Camera port is inserted on the supraumbilical area. Suprapubic 5 mm port is used as an assistant port.

**Figure 3 jcm-11-05651-f003:**
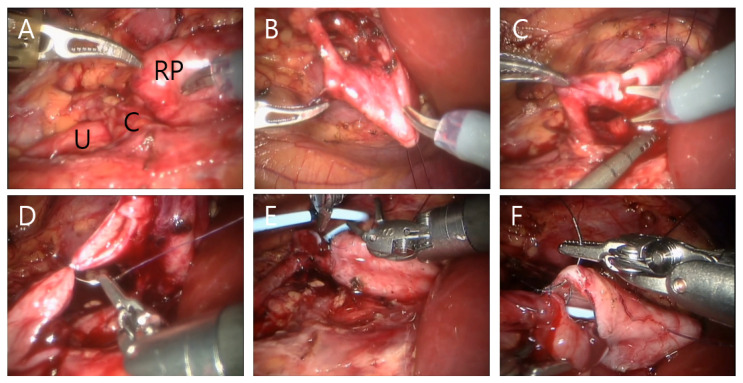
Surgical procedure of right RALP with crossing vessel. Renal pelvis (‘RP’), ureter (‘U’) and crossing vessel (‘C’) were observed (**A**). An anchoring stitch of renal pelvis is being made using 2-0 nylon suture (**B**). Dissection of renal pelvis and longitudinal ureter incision using scissors (**C**). Ureter anastomosis between renal pelvis and ureter was performed using 5-0 polyglyconate and 5-0 or polygalactin suture (**D**). Antegrade DJ stent was indwelled (**E**). Anterior wall anastomosis (**F**).

**Figure 4 jcm-11-05651-f004:**
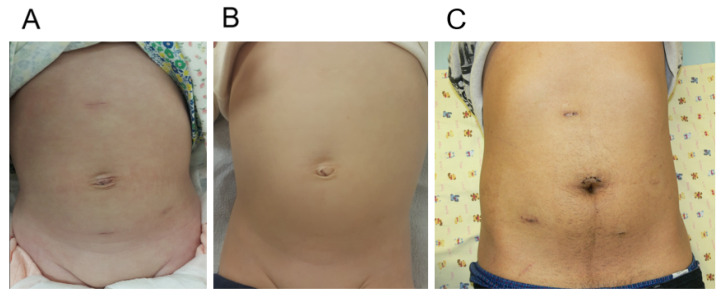
Cosmetic outcomes of robot-assisted laparoscopic pyeloplasty. One month after surgery in infant (**A**) and one year after surgery in infant (**B**). One month after surgery in adult (**C**).

**Table 1 jcm-11-05651-t001:** Patient demographics according to age groups.

	Pediatric Group (*n* = 48)	Adult Group (*n* = 69)
Age at surgery, med (range), (year)	7.5 (0–18)	39.3 (19–74)
Height at surgery, med (range), (cm)	122.5 (62.4–183.7)	166.0 (149.3–186.8)
Weight at surgery, med (range), (kg)	32.4 (7.1–110.3)	63.3 (46.8–96.0)
Body mass index, med (range), (kg/m^2^)	18.7 (12.8–35.6)	22.9 (16.7–31.1)
* Obesity at surgery (%)	13 (27.1)	15 (21.7)
Gender (male:female)	34:14 (70.8:29.2)	31:38 (44.9:55.1)
Laterality (right:left)	13:35 (27.1:72.9)	28:41 (40.6:59.4)
s-Creatinine at surgery, med (range), (mg/dL)	0.5 (0.2–1.5)	0.9 (0.5–1.6)
SFU grade at surgery, *n* (%)		
1–2	2 (4.2)	9 (13.0)
3–4	46 (95.8)	60 (87.0)
APPD at surgery, med (range), (cm)	3.5 (0.7–6.7)	4.1 (1.0–9.4)
SRF at surgery, med (range), (%)	41.8 (5.5–56.6)	36.5 (6.4–58.0)
No. presentation (%)		
Prenatally detected	17 (35.4)	0
Incidentally detected	3 (6.3)	19 (27.5)
Flank/abdominal pain	20 (41.7)	47 (68.1)
Gross hematuria	5 (10.4)	1 (1.4)
UTI	3 (6.3)	2 (2.9)
Pyeloplasty method (%)		
Dismembered	45 (93.8)	69 (100)
Nondismembered	3 (6.3)	0 (0)
Surgical Approach (%)		
Transmesenteric	22 (45.8)	16 (23.2)
Nontransmesenteric	26 (54.2)	53 (76.8)

Continuous and categorical variables were expressed as means ± standard deviations and *n* (%), respectively. * Obesity = either one’s weight percentiles by height > 95 (in children) or BMI > 25 (in adult); s-Creatinine, serum creatinine; SFU, Society for Fetal Urology; APPD, anteroposterior pelvic diameter; SRF, split renal function; UTI, urinary tract infection.

**Table 2 jcm-11-05651-t002:** Perioperative and postoperative outcomes according to age.

	Pediatric Group (*n* = 48)	Adult Group (*n* = 69)	*p*-Value
Total operative time, median (range), (min)	171 (70–324)	148 (65–370)	0.030
Console time, median (range), (min)	126 (78–220)	110 (86–170)	0.020
Anastomosis time, median (range), (min)	63 (15–100)	45 (32–65)	0.271
Etiology, *n* (%)			0.169
Intrinsic-primary	29 (60.4)	34 (49.3)	
Intrinsic-polyp	6 (12.5)	5 (7.2)	
Crossing vessel	13 (27.1)	30 (43.5)	
Postoperative SRF, median (range), (%)	41.7 (11.6–57.5)	38.9 (12.0–59.8)	0.418
Postoperative s-Creatinine, median (range), (mg/dL)	0.5 (0.2–1.1)	0.8 (0.5–1.6)	0.859
Hospital day, median (range), (day)	3.7 (2–12)	4.7 (3–13)	0.278
Pain score ≥ 4 requires analgesics (%)			
Postop day 0	17 (35.4)	46 (66.7)	0.648
Postop day 1	4 (8.3)	23 (33.3)	<0.001
Postop day 2	1 (2.1)	4 (5.8)	0.049
Morphine dose, median (range), (mg/kg)			
Postop day 0	0.08 (0–0.25)	0.15 (0–0.27)	0.005
Postop day 1	0.08 (0–0.33)	0.24 (0–0.40)	0.007
Postop day 2	0.03 (0–0.30)	0.11 (0–0.46)	0.024

SRF, spilt renal function.

**Table 3 jcm-11-05651-t003:** Comparison of the surgical complication outcomes according to age.

	Pediatric Group (*n* = 48)	Adult Group (*n* = 69)	*p*-Value
Surgical complication *	3	2	0.688
Postoperative DJ insertion due to obstruction	1	0	
Ureteroscopic DJ removal due to DJ malposition	1	0	
DJ reposition due to DJ malposition at POD#1	1	0	
Idiopathic hypoxia after surgery	0	1	
RIRS for recurred renal stone removal	0	1	
Secondary procedures needed	3	3	0.688
Surgical failure (%)	2 (4.2)	2 (2.9)	>0.999
Redo pyeloplasty (%)	0 (0.0)	1 (1.4)	
Aggravation of hydronephrosis (%)	1 (2.1)	1 (1.4)	
Decrease in SRF in 36 months (%)	1 (2.1)	0 (0.0)	
Urolithiasis after pyeloplasty (%)	3 (6.3)	2 (2.9)	0.400
ESWL (%)	2 (4.2)	1 (1.4)	
RIRS (%)	1 (2.1)	1 (1.4)	
Urinary tract infection within 30 days after surgery	3	0	0.066

* Clavien–Dindo classification grade ≥3; DJ, double J catheter; POD#1, postoperative day 1; SRF, spilt renal function; ESWL, extracorporeal shock wave lithotripsy; RIRS, retrograde intrarenal surgery.

**Table 4 jcm-11-05651-t004:** Logistic regression analysis of the risk factors for surgical failure.

	Univariate			Multivariable		
	Odds	95% CI	*p*-Value	Odds	95% CI	*p*-Value
* Obesity	10.6	1.1–106.0	0.045	*	*	0.104
Preop stone	14.8	1.5–150.5	0.022	14.5	1.4–147.3	0.022
** Age (pediatric = 1)	1.5	0.2–10.7	0.712			
*** Preop APPD	1.0	0.5–1.8	0.965			
SRF (<30%)	0.9	0.09–9.4	0.960			

* Obesity = either one’s weight percentiles by height > 95 (in children) or BMI > 25 (in adults). ** The pediatric group (age ≤ 18) was used as a reference value. *** Anteroposterior pelvic diameter was processed as a continuous variable. APPD, anteroposterior pelvic diameter; SRF, spilt renal function; CI, confidence interval.

## Data Availability

Not applicable.

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
