# Peer review of "A Comparative Analysis of Robot-Assisted Laparoscopic Pyeloplasty in Pediatric and Adult Patients: Does Age Matter?"

_jcm, 2022, doi:10.3390/jcm11195651_

Round 1

Reviewer 1 Report

This is a well conducted study of interest to surgeons using the robot-assisted instruments.

I noted that the method of operation was trans peritoneal. Why did the authors abandon the retro peritoneal approach?.

The list of references is as expected. The conclusion i  supported by the research data. The presentation is clear and easy to follow.

Where there any conversions or any open surgical procedures conducted during the time of the study?

Author Response

This is a well conducted study of interest to surgeons using the robot-assisted instruments.

Q1. I noted that the method of operation was trans peritoneal. Why did the authors abandon the retro peritoneal approach?.

A1. The approach to the UPJ area can be either transperitoneal or retroperitoneal. We choose the transperitoneal approach because of the surgeon’s preference. We believe that the transperitoneal approach is especially better to identify the crossing vessel around the UPJ area and to freely mobilize the UPJ from the compressing and crossing aberrant vessel. That is because the vessel usually compress the UPJ anteriorly and the transperitoneal approach can provide wider view along the course of the ureter.

Q2. The list of references is as expected. The conclusion i  supported by the research data. The presentation is clear and easy to follow.

A2. Thank you very much for your comment.

Q3. Where there any conversions or any open surgical procedures conducted during the time of the study?

A3. There was no conversion to open or laparoscopic surgery during the study period. There were other cases than robotic surgery that we preplanned and executed the conventional laparoscopic or open pyeloplasty according to the patient’s or parents’ preference or the surgical complexity such as redo-pyeloplasty cases.    

Reviewer 2 Report

It is well written, and detailed demonstration of diverse contents for robotic surgery is impressive in the discussion part.

It seems that several modifications are required.

Authors divided the study group into pediatric (including infant) and adult group. More detailed age distribution may highlight the conclusions of this paper. Is it possible to subdivide the groups more specifically? For example, infant/preschooler/schoolchild/adolescent …

95th line: Is equivalent protocol of perioperative management (e.g. preoperative work up and postoperative follow-up) was applied to both groups? CT scan can be quite helpful in adult group and different (shorter) duration of indwelling catheterization may also be considered, as described in discussion part.

How many surgeons were involved in robotic pyeloplasty? This heterogeneity is important, as surgical skill and learning curve are important factors to obtain ideal outcome.

Using 3 robotic arms can be a solution to shorten operation time and improve surgical outcome.

How about describing surgical tips, especially for infant?

When compared to older pediatric or adult group, smaller port or different robotic arms can be better options to optimize the process of operation. Furthermore, difficulties in surgery may be due to restricted space or fighting of robotic arms.

213th line: The risk of major complication is somewhat higher than that of minor complication. I wonder whether the incidence of minor complication is 7.1% (not 2.9%).

239th line: There were several UTI events after operation. Please, demonstrate the general usage of perioperative prophylactic antibiotics in Asan Medical Center.

In figure 3A, how about making annotations for RP, C, U.

e.g. Renal pelvis (RP), ureter (U) and crossing vessel (C) were observed …

Author Response

Q1. Authors divided the study group into pediatric (including infant) and adult group. More detailed age distribution may highlight the conclusions of this paper. Is it possible to subdivide the groups more specifically? For example, infant/preschooler/schoolchild/adolescent …

A1. In Asan Medical Center, robotic pyeloplasty has been performed since 2013. All patient data from 2013 to the most recent data were collected and documented in this paper. If the data of 117 persons are subdivided by age, the number of samples per group becomes too small, making statistical processing of the data difficult. Nevertheless, we have subdivided the group into the infant, child/adolescent, and the adult group as shown in the supplementary table. When the total number of patients grows enough, we can subdivide the group further as suggested and analyze the data according to the segmented age group.

Supplementary Table. Perioperative and postoperative outcomes according to the age groups.

Infant group

(n = 6)

Child, Adolescent group (n = 42)

Adult group

(n = 69)

P-value

Total operative time, med (range), (mins)

134 (100–210)

176 (70–324)

148 (65–370)

0.024

Console time, med (range), (mins)

87 (60–150)

127 (78–220)

110 (86–170)

0.352

Anastomosis time, med (range), (mins)

70 (60–80)

62 (15–100)

45 (32–65)

0.144

Etiology (%)

0.052

 Intrinsic-primary

6 (100)

23 (54.8)

34 (49.3)

 Intrinsic-polyp

0 (0)

6 (14.3)

5 (7.2)

 Crossing vessel

0 (0)

13 (31.0)

30 (43.5)

Postoperative split renal function, med (range), (%)

45.7 (36.7–55.0)

41.3 (11.6–57.5)

38.9 (12.0–59.8)

0.586

Postoperative s-Creatinine, med (range), (mg/dL)

0.3 (0.2–0.3)

0.6 (0.3–1.1)

0.8 (0.5–1.6)

0.000

Pyeloplasty method (%)

0.064

 Dismembered

0 (0)

3 (7.1)

0 (0)

 Nondismembered

6 (100)

39 (92.9)

69 (100)

Surgical approach (%)

0.018

 Transmesenteric

4 (66.7)

18 (42.9)

16 (23.2)

 Nontransmesenteric

2 (33.3)

24 (57.1)

53 (76.8)

Hospital day, med (range), (day)

3.3 (2–4)

3.8 (2–12)

4.7 (3–13)

0.006

Pain score ≥ 4 requires analgesics (%)

Postop day 0

2 (33.3)

15 (35.7)

46 (66.7)

0.003

Postop day 1

1 (16.7)

3 (7.1)

23 (33.3)

0.000

Postop day 2

1 (16.7)

0 (0)

4 (5.8)

0.001

Morphine dose, med (range), (mg/kg)

Post op day 0

0.05 (0–0.13)

0.08 (0–0.25)

0.15 (0–0.27)

0.000

Postop day 1

0 (0)

0.09 (0–0.33)

0.24 (0–0.40)

0.000

Postop day 2

0 (0)

0.04 (0–0.30)

0.11 (0–0.46)

0.000

*Complications ≥ G3

0

3

3

Secondary procedures needed

0

3

3

Surgical failure (%)

0.784

Redo

0 (0)

0 (0)

1 (1.4)

Aggravation of hydronephrosis

0 (0)

1 (2.3)

1 (1.4)

Decrease of split renal function in 36 months

0 (0)

1 (2.3)

0 (0)

Urolithiasis after pyeloplasty (%)

0.639

 ESWL

 RIRS

0 (0)

0 (0)

2 (4.6)

1 (2.3)

1 (1.4)

1 (1.4)

ESWL, extracorporeal shock wave lithotripsy.

Q2. 95th line: Is equivalent protocol of perioperative management (e.g. preoperative work up and postoperative follow-up) was applied to both groups? CT scan can be quite helpful in adult group. And different (shorter) duration of indwelling catheterization may also be considered, as described in discussion part.

A2. We kept equivalent protocols and methodologies for both groups except that CT scans were saved for pediatric patients. Since CT scan might expose pediatric patients to excessive radiation and renal damage (due to the radiocontrast), the renal status (APPD) of pediatric patients was monitored by ultrasound. And in adult patients, we used CT scan, which was the only difference between post-operative follow-up methodology between two groups. Otherwise, patients were managed with the same number of follow-ups and the same follow-up frequency with the same criteria.

As the reviewer commented, we shortened the DJ stent maintenance duration to decrease the rate of UTI occurrence (more than 4wks to less than 2wks).  (242nd line)

Q3. How many surgeons were involved in robotic pyeloplasty? This heterogeneity is important, as surgical skill and learning curve are important factors to obtain an ideal outcome.

A3. SH Song (corresponding author) performed all 117 surgeries. Therefore, the method and quality of each operation seems to be homogenous. However, this study cohort included the initial 20 patients who underwent RALP during the learning curve period of this surgeon. We have commented this issue as one of the limitations of this study.

Q4. Using 3 robotic arms can be a solution to shorten operation time and improve surgical outcomes. How about describing surgical tips, especially for an infant? When compared to the older pediatric or adult group, smaller ports or different robotic arms can be better options to optimize the process of operation. Furthermore, difficulties in surgery may be due to restricted space or the fighting of robotic arms.

A4. When using the DaVinci S system, we can use 5mm robotic ports and arms. It does minimize the incisional scar for the patient. However, the actual necessary working distance is longer for the 5mm robotic arm than the 8mm robotic arm. It is because the joint movement difference. The 8mm robotic arm uses multi-directional hinge joint movement for the wrist movement. The 5mm robotic arm uses goose-neck type multiple joints for the wrist movement. Therefore, it is better for the surgeon to use 8mm robotic arm for smaller or even in infant patients to deal with the limited working space inside the body. The port placement was not so different even in infant. The most important setup was the depth of the robotic ports. The da Vinci robot platform encourages to put the marked remote center of the cannulae at the level of the abdominal wall to minimize the stress to the patient’s abdominal wall tissue. However, when the working space inside the body is very limited as in infant, the cannulae should be pulled out little more than usual to freely move the robot arms inside the body and to locate the remote center mostly out of the body to do so.

Q5. 213th line: The risk of a major complication is somewhat higher than that of minor complications. I wonder whether the incidence of minor complications is 7.1% (not 2.9%).

A5. In general, the Clavien-Dindo grading is used when classifying the severity of post-surgical complications. In the Journal of Urology article we cited, there is no accurate specification. It is believed that there could be a difference in the rates of minor and major complications if the Clavien-Dindo classification is applied strictly.

Q6. 239th line: There were several UTI events after the operation. Please, demonstrate the general usage of perioperative prophylactic antibiotics in Asan Medical Center.

A6. Generally, 1 g Cefoxitin is given (IV) three times a day starting 30 minutes before the surgery as a prophylactic antibiotics and lasts until the first post-operative day. If the urine culture test performed before surgery and gives positive results, specific antibiotics are chosen instead of Cefoxitin.

Q7. In figure 3A, how about making annotations for RP, C, and U?

A7. Thank you for your kind comment. We added annotations in figure 3A.
